# Exploring the Role of Berberine as a Molecular Disruptor in Antimicrobial Strategies

**DOI:** 10.3390/ph18070947

**Published:** 2025-06-24

**Authors:** Anna Duda-Madej, Szymon Viscardi, Hanna Bazan, Jakub Sobieraj

**Affiliations:** 1Department of Microbiology, Faculty of Medicine, Wroclaw Medical University, Chałubińskiego 4, 50-368 Wrocław, Poland; 2Faculty of Medicine, Wroclaw Medical University, Ludwika Pasteura 1, 50-367 Wrocław, Poland; szymon.viscardi@student.umw.edu.pl (S.V.); hanna.bazan@student.umw.edu.pl (H.B.); jakub.sobieraj@student.umw.edu.pl (J.S.)

**Keywords:** berberine, β-lactamase inhibitors, efflux pump inhibitors, FtsZ protein, isoquinoline alkaloid, multidrug resistance, quorum sensing inhibitors

## Abstract

In recent years, one of the most important issues in public health is the rapid growth of antibiotic resistance among pathogens. Multidrug-resistant (MDR) strains (mainly *Enterobacteriaceae* and non-fermenting bacilli) cause severe infections, against which commonly used pharmaceuticals are ineffective. Therefore, there is an urgent need for new treatment options and drugs with innovative mechanisms of action. Natural compounds, especially alkaloids, are showing promising potential in this area. This review focuses on the ability of the isoquinoline alkaloid berberine (BRB) to overcome various resistance mechanisms against conventional antimicrobial agents. BRB has demonstrated significant activity in inhibiting efflux pumps of the RND (Resistance-Nodulation-Cell Division) family, such as MexAB-OprM (*P. aeruginosa*) and AdeABC (*A. baumannii*). Moreover, BRB was able to decrease quorum sensing activity in both Gram-positive and Gram-negative pathogens, resulting in reduced biofilm formation and lower bacterial virulence. Additionally, BRB has been identified as a potential inhibitor of FtsZ, a key protein responsible for bacterial cell division. Particularly noteworthy, though requiring further investigation, are reports suggesting that BRB might inhibit β-lactamase enzymes, including NDM, AmpC, and ESβL types. The pleiotropic antibacterial actions of BRB, distinct from the mechanisms of traditional antibiotics, offer hope for breaking bacterial resistance. However, more extensive studies, especially in vivo, are necessary to fully evaluate the clinical potential of BRB and determine its practical applicability in combating antibiotic-resistant infections.

## 1. Introduction

Antibiotic resistance among pathogenic bacteria is the “silent pandemic of the 21st century” and is a significant public health problem worldwide. It can result from (i) overuse of antibiotics [1], (ii) incorrect use of antibiotics (taking inappropriate doses or too short a duration of therapy) [2], (iii) improper disposal of antimicrobial drugs [3], and (iv) agricultural overuse [4].

According to the WHO (World Health Organization), an estimated 1.27 million deaths worldwide each year are directly caused by infections with MDR (multidrug-resistant) bacteria, and 4.95 million deaths are indirectly linked to them [5]. Statistical predictions for the near future are very alarming, as they show that, with such an increasing trend, these cases will increase 10-fold in 2050, making antibiotic resistance the number one cause of all deaths in the world, overtaking even cancer [6].

The mechanism of activity of the antibiotics known so far includes their selective action based on interference with key life processes of bacteria, different from those occurring in eukaryotic cells. Antibiotics can act via (i) inhibiting cell wall synthesis (β-lactams and glycopeptides) [7]; (ii) inhibiting protein synthesis (tetracyclines, aminoglycosides, macrolides, lincosamides, streptogramins, oxazolidinones, and chloramphenicol) [8]; (iii) inhibiting DNA/RNA synthesis (quinolones, fluoroquinolones, rifampicin, and metronidazole) [9,10]; (iv) acting as antimetabolites (sulfonamides, trimethoprim, and cotrimoxazole) [11]; and (v) cytoplasmic membrane damage (daptomycin and polymyxins) [12]. Although the field of antibiotic therapy is developing rapidly, the number of antimicrobial drugs discovered in the last half-century is negligible. There is a feeling of a constant deficit of them, especially those active against resistant strains. Indeed, it is a fact that, in recent years, compounds have been discovered promising for the future in combating infections with the etiology of resistant strains, i.e., teixobactin (discovered in 2015) [13], darobactin (discovered in 2019) [14], and lariocidin (discovered in 2025) [15]. However, despite the intense scientific efforts in the field of finding new compounds that exhibit antimicrobial activity, the rate is still far too slow to overcome the development of bacterial resistance to antibiotics.

Over the past decades, bacteria have “learned” a defense against the effective action of drugs. This involves (i) inactivation of the antibiotic [16], (ii) changing its target site of action (overproduction of molecules constituting the target site or replacing them with alternative structures) [17], (iii) reducing the permeability of cellular envelopes [18,19], and (iv) active efflux of the drug from the cell [20]. These modifications can occur as a result of genetic mutations (changes in target site proteins) [21,22] or the ability to produce enzymes hydrolyzing or modifying the antibiotic [23,24]. These comprehensive abilities acquired by bacteria have contributed to the development of resistance to most (MDR), XDR (extensively drug-resistant), and even all (PDR (pandrug-resistant)) antibiotics used in therapy and considered as “last chance” drugs [25]. Additionally, the problem is “me too drugs,” i.e., the fact that most “new” antibiotics were created by modifying existing drugs [26].

Therefore, it is essential to explore novel compounds that exert their effects through mechanisms distinct from those of the currently available agents. A critical aspect of this search involves uncovering new pathways involved in bacterial cell–eukaryotic cell interactions. Furthermore, a comprehensive understanding of the intricate resistance mechanisms employed by various bacterial species is fundamental to the success in this search, as circumventing these defenses would enable the therapeutic agent to act more effectively.

Recent progress in research allowed the development of new drugs, invented in order to overcome bacterial mechanisms of antibiotic resistance. Fabimycin can serve as an example. It inhibits FabI, a rate-determining step catalyzing enzyme, which is involved in fatty acid synthesis. Its ability to accumulate inside targeted bacteria facilitates its effectiveness against resistant Gram-negative bacteria, which was proven in a mouse infection model [27,28]. Another promising compound is INF55 (5-nitro-2-phenyl-1*H*-indole), which not only blocks the pumping of antibiotics out of the bacterial cell (NorA inhibitor) but also increases the efficacy of fluoroquinolones (e.g., ciprofloxacin) [29,30]. Although it is not a stand-alone antibiotic, this compound appears to be the key that opens up a new strategy in the fight against multidrug-resistant bacteria. Acting as a resistance-modifying agent (RMA), it restores the efficacy of existing antibiotics. Notably, research has demonstrated that coupling an efflux pump inhibitor with an antibacterial agent is a highly promising approach [29]. The field of phytomedicine has gained increasing attention, with natural compounds being investigated as potential sources of new antimicrobial agents. One such development is the conjugation of berberine (BRB) with INF55, which has resulted in a hybrid molecule exhibiting potent antimicrobial activity, particularly against *Enterococcus faecalis* and methicillin-resistant *Staphylococcus aureus* (MRSA) [31,32,33].

In our review, we focused on summarizing reports flowing from research on the isoquinoline alkaloid BRB (Figure 1), which is widely present in plants. This compound is present in significant amounts in representatives of the *Berberidaceae,* including *Berberis aristata* or *Ranunculaceae*—*Coptis chinensis* [34,35]. This alkaloid has long held a prominent position among bioactive phytochemicals in traditional and natural medicine due to its broad therapeutic potential. Numerous studies have emphasized its anticancer [36], antimicrobial [37,38], and anti-inflammatory [39] properties. At the same time, BRB, due to its weak absorption from the gastrointestinal tract and high toxic dose (41.6 g/kg), is a safe compound for clinical use [40] and particularly worthy of attention. The reviews available in the literature focus primarily on the very important topic of overcoming BRB bioavailability. They extensively discuss its metabolic profile [41] and physicochemical, pharmacological, and toxicological properties [42]. They focus on the molecular action targets of this compound [43] and the impact of structural modifications to increase therapeutic effects, including improved bioavailability and absorption [44]. In addition, they discuss in detail the positive and negative sides of the formulation of BRB and its derivatives [45]. However, they do not analyze in detail the antimicrobial activity of BRB and the attractive combinations that potentiate DDIs (drug–drug interactions). After analyzing these reviews, there was a need to encapsulate the subject into a whole expanding the area of the effect of this isoquinoline alkaloid on the bacterial cell. Therefore, the main objective of this review is to provide a comprehensive overview of the antimicrobial mechanisms of action of this compound, which positions it as a promising candidate for overcoming the growing challenge of antibiotic resistance. These mechanisms include the disruption of efflux pump function, inhibition of bacterial cell division, interference with quorum sensing pathways, and suppression of β-lactamase activity. They are interesting from the point of view of antimicrobial activity and, in our opinion, worth special attention. This is because it seems that solving the problem of bioavailability and weak permeability of BRB using nanoparticles, phytosomes, or BRB derivatives is the key to overcoming multidrug resistance.

## 2. Berberine Disrupting the Efficiency of Bacterial Efflux Pumps

### 2.1. Berberine Interference with MFS Efflux Pumps Superfamily

According to reports of numerous researchers, BRB as a cationic compound undergoes efficient efflux by MDR bacteria due to the MFS (major facilitator superfamily) efflux pump system, such as NorA [32,46,47]. The result is limited activity against these bacteria and the need for classic EPIs (efflux pump inhibitors), for example, CCCP (carbonyl cyanide 3-chlorophenylhydrazone), PaβN (Phe-Arg-β-naphthylamide), or others newly synthesized, to obtain the antibacterial effect of the alkaloid [30,32,33,48]. Ettefagh et al. also proved that the plant extract from *Hydrastis canadensis* L. can overcome the efflux of BRB conditioned by the presence of NorA in *S. aureus* [49]. In an analysis of 13 alkaloids with antibacterial potential, BRB showed the best antimicrobial profile against MRSA isolates expressing NorA and MdeA pumps (MIC = 125 μg/mL). However, the EtBr (ethidium bromide) analysis did not demonstrate the ability to interfere with the efflux pump system; no accumulation of bromide was observed [50].

However, in recent years, research indicating the ability of the alkaloid to interfere with the function of MFS effluent pumps has emerged [51,52]. Reports on this subject come from the study of Li et al., in which researchers reported on the ability of the preparation in the sub-inhibitory concentration to interfere with the function of the MdfA pump present in *Escherichia coli* [51]. For the strain overexpressing MdfA pump, an 8-fold higher MIC value for ciprofloxacin (CYP) was determined compared to the reference strain. The addition of BRB (20–50 μg/mL) resulted in the restoration of antibiotic susceptibility and the accumulation of fluoroquinolone within *E. coli* cells (decrease in MIC from 0.032 µg/mL to 0.008 and 0.004 μg/mL, respectively). At the molecular level, BRB particles have been shown to interfere with MdfA pump subunits and consequently prevented their conformational transition, which is likely to be the proper mechanism of action of the alkaloid [51]. BRB activity against efflux pump expressing MRSA isolates was also evaluated. The alkaloid showed a synergistic interaction with antibiotics: CYP, tobramycin (TOB), and oxacillin (FICI, fractional inhibitory concentration index ≤ 0.5). Moreover, the time–kill curve analysis showed that, after 24 h exposure to BRB (64 µg/mL) combined with TOB (256 µg/mL) or CYP (0.25 µg/mL), there was 99.9% inhibition of bacterial growth. In the assessment of EtBr accumulation inside *S. aureus* cells, exposure to BRB (32–64 µg/mL) resulted in a fluorescence increase, giving a comparable result to that obtained in the positive control (CCCP, 5 µM). In the docking study, alkaloid presented a significant affinity for efflux pump proteins: MepA, NorA, NorB, and SdrM (binding energy: 7.9, −7.6, −7.8, and −9.4 kcal/mol, respectively) [52].

### 2.2. Berberine Interference with RND Efflux Pumps Superfamily

***Acinetobacter baumannii****:* BRB hydrochloride has been tested by Li et al. for its ability to overcome antibiotic resistance in MDR *A. baumannii* [51]. Importantly, BRB (MIC = 256 μg/mL) in combination with classical drugs led to a restoration of bacterial susceptibility to CYP, tigecycline (TGC), meropenem (MER), and sulbactam (SBT) among MDR isolates originally resistant to them (MIC reduction of antibiotics, respectively: 32-fold, 2-fold, 64-fold, and 16-fold). Importantly, the researchers showed that nearly 100% of the tested strains were characterized by the expression of the AdeABC efflux pump. In the analysis of *adeB* gene expression at BRB stimulation, an upregulation in gene expression was noted in contrast to strains not exposed to the alkaloid. This information, together with the relatively high (−7.42 kcal/mol) binding energy of BRB with AdeB, indicated that the alkaloid acts as a substrate rather than a pure pump inhibitor. Thus, BRB enhanced the effect of antibiotics by competing with them for a binding site in the efflux pump area [51]. Ahmadi et al. showed that the exposure of *A. baumannii* MDR isolates to, among others, BRB and a BRB + thioridazine combination decreased the expression level of the *adeB* gene encoding the subunit of efflux pump AdeABC [53].

***Pseudomonas aeruginosa****:* Morita et al. evaluated the efficacy of BRB in their ability to antagonize aminoglycoside resistance in *P. aeruginosa* expressing the MexXY-OprM efflux pump [54]. Alkaloid (250 µg/mL) led to a decrease in MIC for amikacin (ACN), TOB, and gentamicin (GEN) in the two- to eight-fold range. Importantly, the tested isolates were resistant to all tested aminoglycosides, and a similar relationship in terms of MIC was not demonstrated in the case of pathogens lacking expression of the efflux pumps. BRB, in combination with GEN, has been shown to induce a synergistic relationship (FICI ≤ 0.5) against *P. aeruginosa* harboring a MexXY-OprM pump. In addition, there was a similarly spectacular decrease in MIC of the mentioned antibiotics after a combination with BRB against *Achromobacter xylosoxidans* and *Burkholderia cepacia* (for *A. xylosoxindans*, there was a MIC reduction from 512 to 16 µg/mL for TOB) [54]. Further analysis showed that the alkaloid induced in *P. aeruginosa* wild type (expression of MexAB, MexCD, MexEF, MexXY, and MexVW) a decrease in MIC for erythromycin (ERT) from 256 to 64 µg/mL (four-fold) and TET (tetracycline) from 32 to 16 g/mL (two-fold). Importantly, the knockout strain that preserved only the expression of the MexXY pump was only slightly more sensitive to ERY (eight-fold MIC reduction), which indicated the key role of BRB in blocking this particular pump [54]. The ability of the alkaloid to interfere with the function of the RND-type efflux pump was also assessed by Aghayan et al. Researchers evaluated the effect of BRB on *P. aeruginosa* isolates derived from burn wounds. Among 60 isolated pathogens, ~90% resistant isolates were significantly resistant to ceftriaxone, clindamycin, kanamycin, CYP, and rifampicin (RIF). Importantly, it was shown that the use of a combination of BRB + CYP resulted in a decrease in MIC and MBC parameters for fluoroquinolone, which, taking into account reports from the Morita et al. study, may indicate an inhibition of efflux pumps. This is reflected in the distribution of genes encoding efflux pumps of the tested isolates: ~98% was characterized by the expression of *mexA*, -*D*, and -*F*; *mexX* was found in 96.6% of isolates [55]. The summary of the BRB impact on RND-type efflux pumps is presented in Figure 2.

Significant reports in the field of the anti-efflux properties of alkaloid come from the Su et al. study, which assessed the effect of the BRB + imipenem (IMI) combination against the *P. aeruginosa* (PA012) isolate forming the MexXY-OprM pump. The combination of compounds induced a synergistic relationship (FICI ≤ 0.5) and resulted in a decrease in MIC for both compounds: four- and eight-fold, respectively. Moreover, the use of BRB and IMI in sub-inhibitory concentrations (¼ MIC for BRB) resulted in the accumulation of alkaloid within *Pseudomonas* cells. RT-qPCR analysis showed a decrease in the expression of the genes *mexX*, *mexY*, *mexZ* (~40% reduction), and *oprM* (~50% reduction) [56]. The alkaloid also demonstrated the ability to bind with the MexY protein—a component of the RND pump described above, created by *P. aeruginosa*. In silico, BRB and the aminoglycoside-TOB showed affinity for the antibiotic binding site (antibiotic site). The mentioned alkaloid was characterized by a higher affinity to protein molecules (−864 kJ/mol vs. −454 kJ/mol). The result of the study suggested the formation of a stable MexY-BRB complex protecting TOB from efflux. Moreover, in the subsequent in vitro analysis (30 isolates of *P. aeruginosa*, 100% of which harbored the *mexY* gene), a synergy between BRB (80 µg/mL) and TOB was demonstrated in 12 cases, which was manifested as a reduction in the MIC of the antibiotic ≥ two-fold [57]. Another study investigated the effects of BRB on efflux pumps MexXY-OprM and MexAB-OprM present in *P. aeruginosa*. The use of 200 µM BRB in combination with EtBr resulted in increased accumulation of this in *Pseudomonas* strains harboring MexXY-OprM (inhibition of the efflux ratio equal to 55%). However, the alkaloid did not show any significant effect on the isolate forming the second of mentioned efflux pumps [58]. BRB also demonstrated efficacy against the aminoglycoside-resistant *P. aeruginosa* strain harboring MexXY-OprM. Alkaloid, in a series of studies, has proven to be a candidate for an inhibitor of the protein MexY [59,60]. In the context of the pump, BRB and a number of its aromatic derivatives have been shown to be able to block the allosteric subunit of the MexY protein. A docking study demonstrated the ability of the alkaloid to bind with the described protein (−8.66 kcal/mol) present in the *P. aeruginosa* PA07 strain. Evaluation of a further five clinical isolates of *Pseudomonas* confirmed the beneficial alkaloid profile (docking study) in the dysfunction of the MexXY-OprM efflux pump [60].

***Enterobacteriaceae****:* The ability of BRB to potentiate CYP activity against *Klebsiella pneumoniae* expressing the AcrAB-tolC pump was also evaluated. The researchers showed that the combination of an alkaloid (MIC ≥ 512 µg/mL) with CYP (MIC in the range of 4–512 μg/mL) induced a relationship defined in ~80% of the strains as additive. Interestingly, it was shown that, despite the ability of the alkaloid to increase the antibacterial activity of CYP, this combination of antibiotic (¼ MIC) with BRB (1024 μg/mL) induced a 14-fold increase in the expression of *acrA*, *acrB*, *tolC,* and *acrR* genes. The antimicrobial effect was likely achieved by the interaction with pump proteins or by accumulation of AcrAB in the bacterial cytoplasm, which induced cellular toxicity [61]. These reports are in conjunction with the results of the transcriptome analysis of the *Escherichia coli* K12 strain exposed to BRB. Bacteria under the influence of alkaloid increased the expression of a number of genes responsible for efflux: *acrF* (17-fold increase) and *yhdX-Z* (14–34-fold) [62]. In turn, Gokgoz et al. revealed that *E. coli* K12 exposure to BRB induced a decrease in the expression of genes encoding external membrane proteins, e.g., *ompC*, *ompF,* and *ompW*. The latter protein was described to play an important role in the development of bacterial resistance to various antibiotics [63]. The alkaloid was also effective in interfering with the resistance to COL mediated by MCR-1 (mobilized colistin resistance 1), and in disrupting the AcrAB-TolC efflux pump in *Salmonella* spp. and *E. coli* isolates. Despite the lack of significant antimicrobial activity of BRB (MIC in the range of 625–1250 µg/mL), the combination of EDTA (ethylenediaminetetraacetic acid) and alkaloid (both ¼ MIC) resulted in a decrease in MIC of COL-NS (colistin—non-susceptible) isolates (1–256-fold). The demonstrated ability to lower the MIC for COL in the range of 8–2048-fold reduction was defined as a synergistic interaction for both adjuvants. The BRB docking assay demonstrated the ability of the alkaloid to form complexes with the MCR-1 protein and RND pump subunits AcrB and TolC. Moreover, both BRB and EDTA showed the ability to reduce the expression of *mcr-1* and *tolC* genes [64].

### 2.3. Berberine Interference with Mycobacterium spp. Efflux Pumps

BRB also showed the potential to increase the activity of clarithromycin (CLA) against *Mycobacterium avium intracellulare* (MAC) complex isolates. The preparation, like other EPIs (CCCP, piperine, and tetrandrine), led to an improvement in mycobacteria sensitivity to CLA (in 50% of strains that were originally resistant). The alkaloid in the sub-MIC concentration (20–30 μg/mL) led to ≥four-fold reduction of MIC in 10/12 MAC isolates, and the interaction was defined as synergy. What is particularly important, the *M. avium* subsp. *hominissuis* 2373 strain obtained significant sensitivity to the antibiotic after combining with BRB, and there was a decrease in MIC from 2048 µg/mL to ≤0.25 µg/mL [65]. Puk et al. reported that the combination of BRB (16–64 µg/mL) with a number of antibiotics resulted in the development of synergistic interactions against *Mycobacterium fortuitum*, *chelonae,* and *peregrinum* (*Mycobacterium* other than *tuberculosis*—MOTT). Both a decrease in MICs for tuberculostatics (RIF, streptomycin, and CLA) and aminoglycoside antibiotics were observed. Taking into account the known EPI properties of the alkaloid, it can be assumed that the synergistic interaction was the effect of disturbing the function of the MOTT efflux pumps [66]. The influence of BRB and other EPIs, e.g., verapamil and CCCP, on MAC isolates expressing efflux pumps type ABC (ATP-binding cassette) was also assessed. The researchers reported that *M. avium* subsp. *hominissuis* ATCC 700898 exposure to CLA most strongly induced the overexpression of pump-encoding genes in the pathogen (ABC, MFS, and RND pumps—MmpL). Analysis with EtBr revealed that BRB (MIC = 128 µg/mL) increased the accumulation of bromide in relation to the studied bacterial isolates. What is particularly important, the combination of BRB and CLA as the only one (among the EPIs tested) resulted in an increase in macrolide activity, which was ultimately defined as synergy (FICI ≤ 0.5) [67].

## 3. Berberine Interference with Quorum Sensing

### 3.1. Quorum Sensing and Quorum Quenching

Quorum sensing (QS) is a mode of intercellular communication that allows the bacterial community to adapt to changing circumstances through modifications of gene expression. It can affect the metabolism of the cells in the community, as well as interactions between the cells and, therefore, i.e., the formation of a biofilm or excretion of common goods, such as extracellular enzymes. As those survival strategies are a rising concern in medicine, allowing bacteria to resist therapy, disrupting QS can be useful in combating infections. Such an approach is called quorum quenching and was proven to be effective in inhibiting the production of virulence factors and biofilm by numerous pathogens [68]. To achieve this goal, various enzymes that inactivate QS signals can be used. For example, lactonase SsoPox-I was tested in *P. aeruginosa* pneumonia in rats, in which, when inhaled, reduced PAO1 lasB virulence gene activity, pyocyanin synthesis, proteolytic activity, and biofilm formation [69]. Analogical activity is studied in the case of BRB, which activity as a QS inhibitor can, among other mechanisms, be used to combat bacterial infections.

### 3.2. Berberine Inhibits Quorum Sensing in Pseudomonas aeruginosa

A study by Aswathanarayan et al. investigated BRB sub-inhibitory concentrations as a potential inhibitor of biofilm formation and QS-mediated phenotypes of *P. aeruginosa,* Gram-negative, rod-shaped bacteria responsible for many cases of severe infections, especially hospital-acquired pneumonia. Two strains were used: *P. aeruginosa* PAO1, which is a wild type, and *P. aeruginosa* PAO1-JP2, a mutant unable to produce acyl-homoserine lactones (AHLs), signaling molecules involved in quorum sensing. In the PAO1 strain, BRB at a 0.625 mg/mL concentration reduced biofilm formation by 71.70% [70]. In the case of the mutant strain, 0.078 mg/mL solution was used, and signaling molecules were added: C4-HSL (*N*-Butanoyl-L-homoserine lactone) in one trial and 3-oxo-C12-HSL (*N*-3-Oxo-Dodecanoyl-L-homoserine lactone) in the other. The reduction in biofilm formation was 65.50% and 60.74%, respectively. Compared to the control trial, it proves that BRB inhibits biofilm formation by *P. aeruginosa* through the inhibition of QS. This alkaloid was also among the bioactive metabolites of halophilic bacteria, which showed the inhibition of QS in a multidrug-resistant *P. aeruginosa* [71]. BRB was also identified using affinity chromatography as an active compound in *Phellodendron amurense* extract in a study by Yi et al., in which MIC against *P. aeruginosa* was established as 1280 µg/mL [72]. In this study, ½ MIC of BRB suppressed biofilm formation by 50.5% and reduced the production of pyocyanin, a virulence factor of *P. aeruginosa* by 45.0%. It also inhibited the production of rhamnolipid by 68.1% and significantly reduced the motility of bacteria by 60.3%. The authors suggested that those effects were due to the inhibition of QS through an interaction with the 2-heptyl-3-hydroxy-4-quinolone system (PQS), especially the PqsA protein, and conducted a molecular docking simulation, which suggested the existence of a stable binding interaction between BRB and PqsA.

Another study showed that BRB in combination with azithromycin suppressed quorum sensing more than both substances separately [73]. QS molecules secreted by *P. aeruginosa,* which decreased significantly, included violacein, lasl, lasR, rhlI, and rhlR. In the same study, the synergic activity of the alkaloid and azithromycin was tested in vivo on mice model of chronic lung infection, with *P. aeruginosa* PA03 as the etiological factor. Three days after infection, compared to the control group, and to groups where tested substances were used separately (in the same concentrations), the combined used revealed in dissected lungs of mice the alleviation of abscesses, hemorrhage, and inflammation with lower levels of proinflammatory cytokines Il-6 and IL-8 and higher levels of anti-inflammatory IL-10. In conclusion, the authors suggested that BRB is a promising synergic agent in treating *P. aeruginosa* infections, with quorum sensing inhibition being among crucial mechanisms of action.

### 3.3. Berberine Inhibition of Quorum Sensing and Biofilm Formation in Salmonella enterica subsp. enterica Serovar typhimurium

In the same study by Aswathanarayan et al., BRB was examined against *S. typhimurium*. In the crystal violet assay, the substance at the sub-MIC level showed significant antibiofilm activity; at 0.019 mg/mL, it reduced the biofilm biomass of *S. typhimurium* by 31.20%. In fluorescence microscopy analysis, the same concentration showed no inhibition in the initial attachment of the bacterial cells but suppressed maturing of the biofilm through the inhibition of extracellular polymeric substance production, which can be mediated by quorum signaling [74]. The adhesion and invasion potential of bacteria in the presence of BRB was also determined using human colon cell line HT29 in a monolayer. The highest concentration used in a test was 0.02 mg/mL. The effect was a reduction in adhesion by 54.68% and invasion by 55.37%. What is important, in the control group, this concentration had no negative effect on the human cells. The authors of the study also highlighted that other phytochemicals under research, punicalagin and carvacrol, reduced the invasion of *S. typhimurium* but not adhesion [39,75], which points to BRB as a more active substance. Aswathanarayan et al. also showed that the investigated alkaloid, as well as synthetic QS inhibitor furanone C30, decreased paralysis due to *S. typhimurium* infection in nematode *Caenorhabditis elegans*. BRB at 0.038 mg/mL decreased paralysis of the nematodes by 65.38%, while furanone C30 had slightly greater activity. In conclusion, BRB by disrupting QS seems to inhibit the virulence of *S. typhimurium*.

### 3.4. Berberine Inhibits Quorum Sensing in Hafnia alvei

*H. alvei* is a Gram-negative, facultatively anaerobic bacterium, vastly abundant in the environment and human feces but rarely causing infections. However, it is able to produce biofilms in which the QS process is crucial. Pang et al. investigated the effect of BRB on biofilm formation and quorum sensing by *H. alvei* through the analysis of gene expression using RT-qPCR and high-performance liquid chromatography (HPLC) [76]. In RT-qPCR, two genes involved in QS and AHL synthesis were investigated and their expression significantly decreased: the halI gene (even at the lowest investigated BRB concentration, 1/32 MIC) and halR gene (at 1/16 MIC and greater concentrations). HPLC revealed a decrease in the concentration of one of the AHLs, C14-HSL, from 2.189 µg/mL to 0.512 µg/mL after treatment with ½ MIC of BRB. Further docking analysis and molecular dynamics simulation showed that both the alkaloid and C14-HSL bind to the HalR protein, with BRB forming a more stable complex. In conclusion, BRB seems to inhibit the biofilm formation of *H. alvei* by inhibiting QS, and the mechanism behind it is the downregulation of *hal*I and *hal*R gene expression.

### 3.5. Berberine Inhibits Quorum Sensing in Escherichia coli

A study by Sun et al. compared the quorum quenching activity of BRB and another substance derived from Chinese traditional medicine, matrine. The effect on antimicrobial-resistant (AMR) *E. coli* biofilm formation and gene expression was examined using laser scanning confocal microscopy (LSCM) and RT-PCR. Analysis showed that BRB at ½ MIC (MIC = 2560 µg/mL for all AMR strains) significantly inhibited biofilm formation and had an impact on the morphology of the bacterial cells: they were swollen and elongated, with yellow fluorescence from BRB [77]. RT-PCR revealed the downregulating effect of the substance on the expression of the genes involved in QS: *luxS*, *pfS*, *hflX*, *ftsQ,* and *ftsE* at ½ and ¼ MIC. Additionally, the results showed greater activity of BRB than matrine.

### 3.6. Berberine and Quorum Sensing in Gram-Positive Bacteria

Among Gram-positive bacteria, *S. aureus* is a species with prominent clinical significance, especially strains resistant to antibiotics, such as MRSA. BRB is among the substances exhibiting potential in the treatment of MRSA. One of its proven properties is the ability to inhibit the biofilm formation of this pathogen. A study by Chu et al. examined this characteristic and suggested it is possible due to the hydrophobic interaction between BRB and the phenyl ring in phenylalanine, one of the amino acids present in phenyl soluble modulins (PSMs), particularly PSMα2, which form amyloid fibrils, constituting biofilm [78]. BRB was also among active compounds found in goldenseal (*Hydrastis canadensis*) leaf extract; however, it had lesser activity alone than the whole extract, probably due to other flavonoids’ (sideroxylin, 8-desmethyl-sideroxylin, and 6-desmethyl-sideroxylin) additional effect [79]. The extract showed quorum quenching activity against MRSA isolates, which was confirmed by assay with fluorescent reporters and with a lux reporter. Researchers focused on accessory gene regulator (*agr*), a QS system responsible for the production of virulence factors in response to external conditions [80]. The study showed the inhibitory effects of *H. candensis* extract on *agr I*, *agr II,* and *agr III*; however, BRB alone at 75 µg/mL had no effect on the *agr* system.

Contradictory data came from a study by El-Hamid et al. in which the BRB effect on vancomycin-resistant *S. aureus* (VRSA) was investigated. The study suggested a complex mechanism of action of the substance, including QS inhibition [81]. Again, the *agr* system was a subject of researchers’ interest. In the described study, the examined strains contained *agr I* and *agr III* alleles. Those VRSA strains exhibited high biofilm formation and virulence potential. In the study, they were exposed to free BRB and BRB-loaded mesoporous silica nanoparticles (MPS-NPs). After exposure, qRT-PCR revealed significantly lower levels of *agr* transcripts: in the case of the free substance, a 0.29-fold decrease and, in the case of MPS-NPs, a 0.11-fold decrease. The same strains were tested in vivo in mice infected with 5 × 10^6^ CFUs subcutaneously and then treated topically with a free alkaloid and MPS-NPs. Both treatments resulted in lower rates of virulence factors and *agr* expression, as well as smaller clinical symptoms (skin lesions vs. abscesses and reduction in mice activity) and lower levels of proinflammatory cytokines. Those results are supported by a study by Gao et al., who examined BRB and amphotericin B synergic activity against biofilm formed by *S. aureus* and *Candida albicans* [82]. The authors analyzed the gene expression of *S. aureus* in the biofilm treated with 128 µg/mL BRB and 4 µg/mL amphotericin B. The mRNA expression of *agrA* underwent a 125-fold decrease. Therefore, BRB seems to act through QS inhibition, particularly the *agr* system, to reduce *S. aureus* virulence.

### 3.7. Summary

BRB acts as a QS inhibitor at sub-MIC concentrations. This property was proven for multiple bacterial species: *P. aeruginosa*, *S. enterica*, *S. aureus*, *E. coli,* and *H. alvei*. The molecular mechanisms involved are presented in Table 1. Possible effects of this activity include disruptions in biofilm formation and the production of virulence factors, which, in the case of human pathogens, especially antibiotic-resistant strains, can have clinical implications. Therefore, the quorum quenching properties of BRB should be investigated further, as one of the mechanisms responsible for potential application in the treatment of infections.

As it was presented, BRB may interfere with various QS pathways in numerous bacterial species, both Gram-positive and -negative. Biofilm formation is crucial for pathogen survival and resistance against the immune system and is also strictly associated with QS signaling. The mentioned process is presented in Figure 3.

## 4. Berberine—Bacterial Mitosis Inhibitor

### 4.1. Bacterial Division

Bacteria increase their number through mitosis. This process is called division. The whole life of a bacterial cell can be split into three stages marked by the end of division, the beginning of chromosomal replication, and its end [83]. One of the most important proteins partaking in the bacterial mitosis is the FtsZ protein [84]. This tubulin homolog polymerizes into a cycle, commonly called a “Z-ring”, that determines the division site and recruits other necessary proteins. It has been observed that FtsZ polymerization is highly dependent on guanosine-5′-triphosphate (GTP). Understanding of the mechanisms underlying bacterial divisions is crucial in order to come up with new antibiotics or their adjuvants.

### 4.2. Mechanism of Berberine and FtsZ Protein Interaction

BRB’s antibacterial activity can be partly explained by its interaction with the FtsZ protein. One study proved that this alkaloid connects to the GTP-binding pocket present on FtsZ, effectively inhibiting its activity. BRB also prevents FtsZ’s polymerization, as it covers its binding sides. This results in mitosis rate reduction [85]. This thesis has been supported by another study carried out in 2014, in which BRB and its derivatives’ activity against various bacteria, e.g., *S. aureus*, *E. faecium,* or *E. coli*, has been compared [86]. Researchers observed a visible decrease in FtsZ protein’s GTPase activity. GTPase is an enzyme that plays the role of a molecular switch that can control certain cellular processes. This may be a reason behind the observed reduction of the polymerization rate.

### 4.3. Berberine Activity on Gram-Positive Bacteria

One of the available studies used Gram-positive bacteria—*S. aureus* and *E. faecalis*—to verify the antimicrobial activity of BRB’s derivatives in comparison to BRB [87]. Substances were applied both to antibiotic-susceptible (methicillin-susceptible *S. aureus*—MSSA and vancomycin susceptible *E. faecalis*—VSE) and antibiotic-resistant (MRSA and vancomycin-resistant *E. faecalis*—VRE) strains of those bacteria. While commonly used antibiotics, such as TET or oxacillin, demonstrated higher efficacy against MSSA and VSE compared to BRB derivatives, the latter exhibited significantly lower MIC values when tested against antibiotic-resistant strains (MRSA and VRE). The mechanism behind their efficiency has been studied in a 90° angle light scattering assay, which proved that those derivatives stimulate FtsZ polymerization. Such stimulation results in cell death due to the toxicity of the abundance of the FtsZ protein [88]. In another study, multiple derivatives of cycloberberine were synthesized in order to examine their antimicrobial properties [89]. They were tested on both methicillin-susceptible and -resistant strains of *S. aureus*, and their activity was assessed through the agar dilution method. Two compounds proved themselves to be the most effective even against MRSA, and one of them has been shown to interact with the docking side of the FtsZ protein in a computer program simulation, thus inhibiting mitosis. This corresponds to the conclusions achieved in other studies. Research conducted in 2020 tested various BRB derivatives against a few bacteria species, two of them being Gram-positive *S. aureus* and *B. subtilis* [90]. The results showed that some of those compounds are really effective against these bacteria, especially when compared to pure BRB. Like in previous research, even antibiotic-resistant strains have been proven vulnerable against tested BRB derivatives. In vitro biological assays have shown that, once again, these properties may be the result of interactions of those compounds with FtsZ, as they revealed that the BRB core of those substances links with the binding side of the protein. This mechanism results in a decreasing number of cell divisions.

### 4.4. Berberine Activity on Gram-Negative Bacteria

A study conducted in 2022 investigated the effects of berberine chloride on bacteria, including Gram-negative *Salmonella enterica* subsp. *enterica* serovar Typhi [91]. This compound was compared to other potential FtsZ inhibitors and ranked among them as the most effective. Notably, it demonstrated superior—that is, lower—MIC values against Gram-positive bacteria as compared to the Gram-negative ones. In this research, it was proven that berberine chloride prevented polymerization of the FtsZ protein and reduced its activity as a GTPase. This resulted in a decrease in mitosis. Comparable findings were reported in a study involving *Shigella flexneri*, which used berberine chloride as a reagent in both in vitro and in vivo murine models [92]. In vitro, this compound mostly presented an inhibitory effect on the FtsZ protein, thereby reducing its GTPase activity. Its antimicrobial activity against *S. flexneri* aligns with the findings of an in vivo study, which revealed a dual mechanism of action for BRB. It turned out that this substance not only slows down the mitosis rate of bacteria but, in addition, reduces inflammation and pyroptosis. It results in protection against intestinal infection caused by *S. flexneri*. The inhibitory effects of BRB’s derivatives were also assessed on *E. coli*. In one of the studies, those substances were used on a recombinant FtsZ protein retrieved from *E. coli* [93]. Like in previous research, an inhibition of FtsZ polymerization was observed. Similarly to the Naz et al. study, the MIC values against Gram-positive bacteria were lower compared to the Gram-negative ones [91]. In another research, the effectiveness of simplified BRB’s analogs was tested on various bacterial strains, including *E. coli* [94]. The results of this study correspond with the already named ones, as, once again, it turned out that BRB’s derivatives prevent the polymerization of the FtsZ protein and reduce its GTPase activity, leading to a decrease in bacterial cell division. Their antimicrobial activity has also been assessed as higher against Gram-positive bacteria compared to Gram-negative ones.

### 4.5. Other Studies Concerning the Berberine Effect on Bacterial Mitosis

One of the studies has investigated the effect BRB has on *M. tuberculosis*, as it is one of the promising new antitubercular drugs [95]. In correspondence to the previous research, it turned out that BRB’s analogs created stable complexes with the *M. tuberculosis* FtsZ protein, effectively inhibiting these bacterial divisions. Their efficiency appeared to be better compared to pure BRB. This research also explored pharmacokinetics through assessment of the ADMET (Absorption, Distribution, Metabolism, and Excretion) properties of these compounds, predicting their potential as orally administered drugs. BRB’s antimitotic effect on bacteria was also studied on the fission yeast cells, which were genetically modified to express the FtsZ protein derived from, respectively, *S. aureus* and *Helicobacter pylori* [96]. In this research, BRB was compared to other FtsZ protein inhibitors, and its toxicity towards eukaryotic cells was also assessed. The results of the study were in accordance with the previous ones, and the inhibition of bacteria division was observed. It has been noted, though, that BRB has influenced yeast cells physiology, leading to negative side effects. Scientists have also identified a likely cause of BRB resistance in some bacteria, as certain interactions within the FtsZ protein appear to hinder it and other inhibitors from accessing the binding site. While this study confirmed once again the antimicrobial features of BRB, it also shed some light on new challenges related to it.

The mechanism of action described above of the FtsZ protein and its interaction with BRB is presented in Figure 4.

## 5. Berberine—β-Lactamases Inhibitor

### 5.1. β-Lactamases as an Antibiotic Resistance Mechanism

One of the most commonly used groups of antibiotics are β-lactams. Those drugs are divided into four groups: penicillins, cephalosporins, monobactams, and carbapenems [97]. They all have one common structural feature, which is the β-lactam ring. β-lactams use it in order to bind with penicillin-binding proteins (PBPs), which results in the disruption of transpeptidation and thus inhibits the synthesis of bacterial cell walls. Disturbance in this process leads to bacteria cell death. One of the strategies developed by bacteria against β-lactam antibiotics is enzymatic resistance through peptides called β-lactamases [98]. They hydrolyze β-lactam rings, rendering them ineffective. Like β-lactam antibiotics, these enzymes can also be classified into several groups, the most commonly used system being the Ambler classification [99]. It is presented in Table 2 with exemplary β-lactamases.

### 5.2. Berberine as an Antibiotic Adjuvant and β-Lactamases Inhibitor

β-Lactamases pose a threat to modern medicine, as they can inactivate antibiotics used to treat infections. In order to prevent them from doing so, antibiotic adjuvants such as β-lactamase inhibitors e.g., clavulanic acid, are used [102]. Those compounds do not damage bacteria themselves but rather protect antibiotics from degradation. Promising new adjuvants also belong to BRB, which antimicrobial activity has been observed even towards highly antibiotic-resistant strains [103]. One such study evaluated BRB potential as a New Delhi metallo-β-lactamase-1 (NDM-1) inhibitor [104]. Scientists conducted molecular docking tests to assess the binding affinities of selected phyto-ligands to the active site of examined β-lactamase produced by *E. coli*. Analysis indicated that BRB effectively interacted with NDM-1, thus potentially inhibiting its activity. As there are no potent metallo-β-lactamase inhibitors for now [105], those results are quite promising. Another research conducted in 2020 analyzed BRBe’s activity on β-lactamases produced by *M. tuberculosis* (BlaC) [106]. This enzyme belongs to Ambler class A [107]. Constructed models have shown that this alkaloid can bind with the catalytic side of examined β-lactamase, thus inhibiting its hydrolytic properties. Notably, BRB demonstrated a lower bond energy than the referential antibiotic (ceftazidime). This results in a more stable bond with BlaC, thereby increasing the BRB antimicrobial efficacy. BRB’s activity as a potential β-Lactamase inhibitor has also been tested on β-Lactamase retrieved from *S. aureus* [108]. Docking studies against this enzyme have shown that BRBe forms bonds with key amino acid residues in the catalytic center. This may result in an inhibitory effect on this β-Lactamase. It is worth mentioning that, among the tested substances, BRB stood out due to its lowest binding energy. It contributes to the stability and efficiency of its bond with the examined *S. aureus* β-lactamase. A newer study conducted in 2024 focused on BRB interaction with class C β-lactamase (AmpC) [109]. In order to assess its inhibitory activity on AmpC, molecular docking studies have been carried out. They indicated that BRB binds with this β-lactamase with a suitable pose and low binding energy, thus stabilizing the structure. The results seem promising and validate BRB’s position as a potential new β-lactamase inhibitor. In another study, BRB inhibitory properties were tested on extended-spectrum β-lactamase derived from *E. coli* [110]. This alkaloid was compared to various other compounds in a molecular docking test. While it did not emerge as the most effective one, this study nonetheless confirmed BRB’s anti-β-lactamase activity, resulting in its inclusion in pharmacodynamic assessments. Notably, BRB demonstrated the largest zone of inhibition among the tested compounds, although its minimum inhibitory concentration was relatively average compared to the others.

## 6. Materials and Methods

In this review, we searched for articles by using the databases Scopus, PubMed, Web of Science, and Google Scholar. In total, 110 articles were cited. The articles were qualified for review by searching for the following keywords in the title and abstract of the articles: “berberine”, “efflux pump”, “quorum sensing”, “FtsZ”, and “β-lactamase”. This search excluded patent literature. The literature was selected to demonstrate the purpose of BRB on bacteria. In addition, only actives in English were included. Figure 1 shows the chemical formula of alkaloid described in the review. Figure 2 depicts a simplified scheme of the molecular action of berberine on an RND-type efflux pump. Figure 3 presents a simplified sequence of biofilm formation. Table 1 presents a summary of the BRB properties as a quorum sensing inhibitor. Table 2 stands for the Ambler classification of β-lactamases.

## 7. Conclusions

Given that no new class of antimicrobial agents with novel mechanisms of action has been developed over the past two decades, natural compounds—including BRB—have emerged as promising candidates for pharmaceutical research.

In this review, we summarize the current state of knowledge regarding the multiple antimicrobial mechanisms of BRB, which potentially enable it to overcome resistance among high-priority pathogens, e.g., *P. aeruginosa* and *A. baumannii*. Importantly, no effective and safe efflux-targeting antibacterial agents have yet been approved for clinical use, despite the central role of efflux in mediating multidrug resistance in critical pathogens. Thus, BRB appears to be a promising molecule in this area. BRB has demonstrated the ability to inhibit the MexAB-OprM efflux pump, a broad-substrate resistance determinant that contributes to decreased susceptibility to most therapeutic antibiotics. Furthermore, reports indicate that BRB can suppress the expression of the *AdeABC* efflux pump genes in *A. baumannii*.

Notably, BRB has also been shown to interfere with QS, thereby disrupting biofilm formation and antimicrobial resistance development in both Gram-positive and Gram-negative pathogens. By impairing QS signaling, BRB may reduce catheter-related infections, an issue of major concern in the hospital environment. QS inhibitors (QSIs) are regarded as promising antimicrobial candidates due to their ability to interfere with bacterial cell-to-cell communication and prevent infection onset. However, their clinical implementation remains distant and requires further investigation, especially under in vivo conditions.

A similar situation applies to bacterial mitosis inhibitors targeting the FtsZ protein. The unique mechanism by which BRB affects bacterial cell division is of particular interest due to the high conservation of FtsZ, suggesting that berberine-based compounds could potentially act against a broad spectrum of bacterial species.

In the context of a renewed focus on next-generation β-lactamase inhibitors, such as zidebactam, vaborbactam, taniborbactam, and relebactam, natural substances that are not structural analogs of penam have increasingly shown inhibitory activity against these enzymes. Among them, BRB appears especially promising. Due to its broad inhibitory spectrum—targeting ESβLs, AmpC, and NDM—BRB demonstrates a significant potential for the development of new β-lactamase inhibitors aimed at overcoming β-lactam resistance in multidrug-resistant bacterial strains. In particular, its activity against metallo-β-lactamases is of critical interest in light of the growing prevalence of such infections in intensive care units (ICUs) and should motivate further exploration of this compound.

In conclusion, BRB antimicrobial activity involves diverse mechanisms, including the disruption of efflux pumps, suppression of QS and mitosis, and the inhibition of β-lactamases. These properties state its potential as a future treatment option in infections caused by antibiotic-resistant bacteria. However, further research is needed to confirm the efficiency and safety in humans, assess the pharmacological properties, and invent suitable forms for medical use. BRB can also serve as a starting point for the development of even more useful derivatives or compounds with similar activity.

## Figures and Tables

**Figure 1 pharmaceuticals-18-00947-f001:**
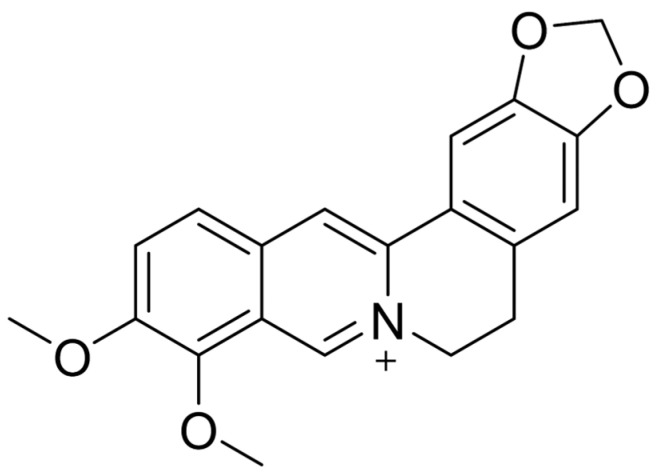
Molecular structure of berberine (two-dimensional).

**Figure 2 pharmaceuticals-18-00947-f002:**
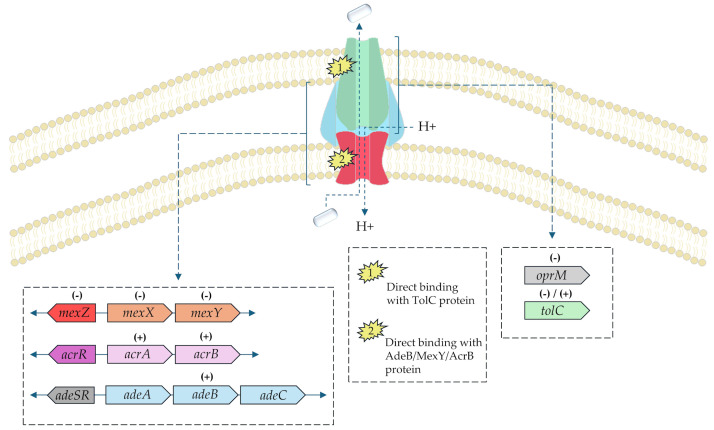
Berberine anti-efflux activity against Gram-negative bacteria expressing RND-type efflux pumps (*P. aeruginosa*, *A. baumannii,* and the *Enterobacteriaceae*). Abbreviations: *acrA,B,R*, *adeA–C,S,R*, *mexX–Z*, *oprM*, and *tolC*—genes encoding efflux pumps subunits and their negative regulators; AcrB, AdeB, MexY, and TolC—efflux pumps protein subunits; H+—proton. Upregulation was marked as (+) and downregulation as (−).

**Figure 3 pharmaceuticals-18-00947-f003:**
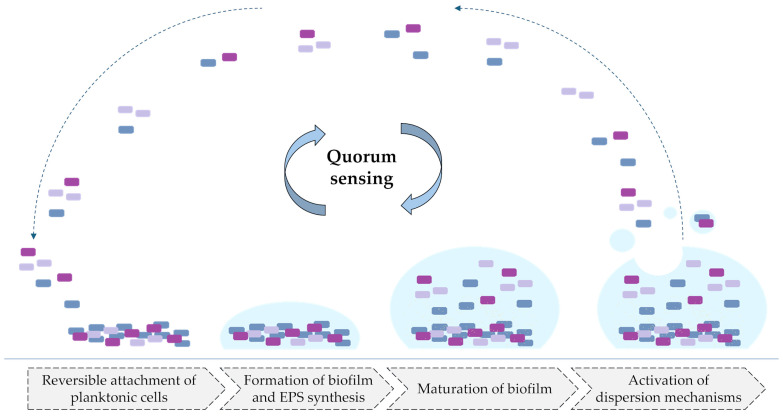
Molecular basis of the quorum sensing role in the process of biofilm formation (simplified scheme). Abbreviations: EPS—exopolysaccharides.

**Figure 4 pharmaceuticals-18-00947-f004:**
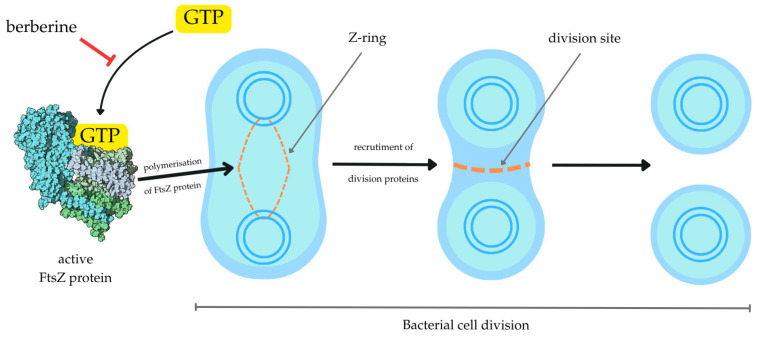
FtsZ protein’s role in bacterial cell division and its interaction with BRB. Abbreviations: GTP—guanosine tri-phosphate.

**Table 1 pharmaceuticals-18-00947-t001:** Molecular effects of berberine on quorum sensing in pathogenic bacterial species.

Bacterial Species	Molecular Effect	References
*P. aeruginosa*	inhibition of PqsA, violacein, *lasl, lasR, rhlI* and *rhlR*	[72,73]
*S. typhimurium*	inhibition of EPS production	[70]
*S. aureus*	inhibition of *agr I, agr II, agr III* systems	[79,81]
*E. coli*	downregulation of *luxS*, *pfS*, *hflX*, *ftsQ* and *ftsE*	[77]
*H. alvei*	reduction of C14-HSL production	[76]

**Table 2 pharmaceuticals-18-00947-t002:** Ambler classification with exemplary enzymes.

Ambler Class	Active Site Type	Examples	References
A	Serine-β-lactamase	KPC, TEM-1	[100]
B	Metallo-β-lactamase	NDM, VIM, IMP	[99]
C	Serine-β-lactamase	AmpC, ADC	[101]
D		OXA-23, OXA-48	[100]

Abbreviations: ADC—*Acinetobacter*-derived cephalosporinase, KPC—*Klebsiella pneumoniae* carbapenemase, TEM-1—narrow spectrum β-lactamase TEM-1, OXA-23—Oxacillinase-23, OXA-48—Oxacillinase-48, NDM—New Delhi metallo-β-lactamase, VIM—Verona integron-encoded metallo-β-lactamase, IMP—Imipenemase.

## Data Availability

No new data were created or analyzed in this study. Data sharing is not applicable to this article.

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
