# Peer review of "Exploring the Role of Berberine as a Molecular Disruptor in Antimicrobial Strategies"

_pharmaceuticals, 2025, doi:10.3390/ph18070947_

Round 1
Reviewer 1 Report
Comments and Suggestions for Authors
This review is focused on the properties of the alkaloid berberine as an antimicrobial able to overcome numerous mechanisms supporting the alarming phenomenon of drug resistance. Literature reviews are in great demand in the field of berberine research, as this alkaloid is multipotent. While promiscuity and ADMET issues seriously bias its clinical usefulness, the study of berberine activities may inspire the research of related compounds to address its defects. The paper is well-structured and generally well-written (apart from some points that should be addressed; see below). While numerous general reviews on berberine can be found, the paper under evaluation presents an original tract in that it is focused on a specific topic: berberine antimicrobial potential. Thus, the review under evaluation is relevant to the Journal and could be accepted provided the authors address the following points of criticism.
- An indication of the time limitation of the review should be added. The author should state that the patent literature was not considered. It would seem that only bacteria were considered as targets: it would be a touch of kindness to specify this limitation. Finally, it would seem that only papers written in English were reviewed.
- The difference between this work and the previously published ones should be highlighted, citing the most recent and relevant general reviews on the pharmacological activities of berberine. Some exemplary papers are the following:
Jamshaid, F.; Dai, J.; Yang, X. L. New Development of Novel Berberine Derivatives against Bacteria. Mini-Reviews in Medicinal Chemistry 2020, 20 (8), 716–724. https://doi.org/10.2174/1389557520666200103115124. This paper is the only review I know that shares the same focus as the review under evaluation: it should be reported and critically compared underlying the points of novelty of the work under evaluation.
Sun, P.; Wang, Z.; Ma, Y.; Liu, Y.; Xue, Y.; Li, Y.; Gao, X.; Wang, Y.; Chu, M. Advance in Identified Targets of Berberine. Frontiers in Pharmacology 2025, Volume 16-2025. https://doi.org/10.3389/fphar.2025.1500511
Karimi, S.; Heidari, F.; Amraei, S. Therapeutic Aspects of Berberine and Its Derivatives: Recent Advances and Challenges. Nano Micro Biosystems 2023, 2 (1), 42–47. https://doi.org/10.22034/nmbj.2023.390449.1018
Kong, Y.; Li, L.; Zhao, L.-G.; Yu, P.; Li, D.-D. A Patent Review of Berberine and Its Derivatives with Various Pharmacological Activities (2016–2020). Expert Opinion on Therapeutic Patents 2022, 32 (2), 211–223. https://doi.org/10.1080/13543776.2021.1974001 (to cover the patent literature).
Filli, M. S.; Ibrahim, A. A.; Kesse, S.; Aquib, M.; Boakye-Yiadom, K. O.; Farooq, M. A.; Raza, F.; Zhang, Y.; Wang, B. Synthetic Berberine Derivatives as Potential New Drugs. Brazilian Journal of Pharmaceutical Sciences 2022, 58. https://doi.org/10.1590/s2175-97902020000318835
Patel, P. A Bird’s Eye View on a Therapeutically ‘Wonder Molecule’: Berberine. Phytomedicine Plus 2021, 1 (3), 100070. https://doi.org/10.1016/j.phyplu.2021.100070 (this wide review considers also the antibacterial properties of berberine)
Wang, K.; Feng, X.; Chai, L.; Cao, S.; Qiu, F. The Metabolism of Berberine and Its Contribution to the Pharmacological Effects. Drug Metabolism Reviews 2017, 49 (2), 139–157. https://doi.org/10.1080/03602532.2017.1306544 (not only does this review include some papers dealing with the antimicrobial activities of berberine, but also discusses the main bioavailability issues that seriously hamper the use of this drug as a medicine).
Wang, J.; Tang, J.; Yu, L.-F.; Li, J.; Li, J.-Y. Berberine Analogues: Progress Towards Versatile Applications. HETEROCYCLES 2015, 91, 2233. https://doi.org/10.3987/REV-15-825.
- The major ADMET issues in berberine as a medicine were not critically addressed.
- The following paragraphs appear as written by AI:
Abstract: The increasing resistance to antibiotics poses a growing threat to public health 12 worldwide. The high prevalence of multidrug-resistant (MDR) strains, particularly 13 among Enterobacteriaceae and non-fermenting bacilli (Pseudomonas aeruginosa, Acinetobacter 14 baumannii) in intensive care units, severely limits treatment options. In response, 15 researchers are prioritizing the search for new generations of pharmaceuticals with 16 unique mechanisms of action
Research in this area is progressing rapidly. In recent years, promising strategies have emerged that specifically address this challenge. Notably, certain compounds have been developed that demonstrate the capacity to overcome traditional bacterial defenses. One such example is fabimycin—a rationally designed antibiotic that targets a key 87 enzyme involved in fatty acid biosynthesis in Gram-negative bacteria. This compound not 88 only exhibits enhanced permeability across the outer membrane but also circumvents 89 resistance mechanisms related to efflux pump activity [27,28].
In conclusion, BRB exhibits a range of antibacterial mechanisms distinct from 622 conventional agents, making it a strong candidate as a lead compound in combating 623 antimicrobial resistance. Further studies focused on its delivery methods and formulation 624 will be crucial in overcoming its potential toxicity to the human host. Such efforts may 625 offer a new therapeutic paradigm for managing infections caused by multidrug-resistant 626 pathogens, while also providing a robust platform for the design of next-generation 627 antimicrobials capable of circumventing existing resistance mechanisms.
They should be rewritten without the help of AI.
- Several grammar or syntactic errors, as well as typos, should be easily amended. An exemplary list is reported below.
Abstract, line 8: the hyphen after ‘alkaloid’ should be left out.
Introduction, page 2, line 13: ‘roduction of’ should be replaced with ‘acting as’.
Introduction, page 2, line 18: ‘interestingly’ is redundant and should be left out.
Introduction, page 2, lines 22-23: the end of the sentence (‘bacteria are ahead of us by several or even a dozen steps’) is not scientifically sound.
Introduction, page 2, line 50: the indicated hydrogen should be typed in italics.
Section 2, page 4, line 1: probably, ‘of’ should be placed between ‘study’ and ‘Li’.
Section 2, page 4, line 11: the article ‘The’ is missing before ‘Alkaloid’.
Section 2, page 4, line 13: a space should separate ‘24’ from ‘h’.
Section 2, page 4, line 18: the sign minus should be given as emdash (three times).
Section 2, page 4, subsection 2.2, line 10: the sign minus should be given as emdash.
Section 2, page 4, subsection 2.2, line 29: ‘g/mL’ or ‘microg/mL’?
Section 2, page 6, subsection Enterobacteriaceae, line 3: I think that ‘the’ should be preferred to ‘an’.
Section 3, subsection 3.1, page 7, line 4: why ‘e.g.’?
Section 3, subsection 3.1, page 7, line 18: the hyphen is not needed.
Section 3, subsection 3.3, page 8, line 15: the name ‘furanone’ should be followed by ‘C30’.
Section 3, subsection 3.4, page 8, line 1: ‘bacteria’ or ‘bacterium’?
Section 3, subsection 3.6, page 9, second paragraph, line 1: ‘contradictive’ or ‘contradictory’?
Section 3, subsection 3.6, page 9, second paragraph, line 1: ‘comes’ or ‘came’?
Section 3, subsection 3.6, page 9, second paragraph, line 15: ‘S. aureus’ should be typed in italics.
Section 3, page 10: the paragraph preceding Fig. 3 is redundant and may be left out.
Figure 3 is out of place and adds nothing: I suggest its removal.
Section 4, subsection 4.1, page 10, line 5: ‘cirque’ or ‘cycle’?
Section 4, subsection 4.3, page 11, line 2: ‘pure’ is redundant.
Section 4, the second paragraph: The captions for the Figures and Tables make no sense there.
Author Response
Dear Reviewer,
Thank you very much for studying the topics undertaken in our Article and for your detailed review. In correcting our Review, we followed all your recommendations, for which we are very thankful. Our corrections are presented below:
- An indication of the time limitation of the review should be added. The author should state that the patent literature was not considered. It would seem that only bacteria were considered as targets: it would be a touch of kindness to specify this limitation. Finally, it would seem that only papers written in English were reviewed.
We would like to thank you for this valuable comment. All this necessary information, previously omitted by us, has been added in the “Materials and Methods” section. They are as follows:
„This search excluded patent literature. The literature was selected to demonstrate the purpose of BRB on bacteria. In addition, only actives in English were included.”
- The difference between this work and the previously published ones should be highlighted, citing the most recent and relevant general reviews on the pharmacological activities of berberine. Some exemplary papers are the following:
Jamshaid, F.; Dai, J.; Yang, X. L. New Development of Novel Berberine Derivatives against Bacteria. Mini-Reviews in Medicinal Chemistry 2020, 20 (8), 716–724. https://doi.org/10.2174/1389557520666200103115124. This paper is the only review I know that shares the same focus as the review under evaluation: it should be reported and critically compared underlying the points of novelty of the work under evaluation.
Sun, P.; Wang, Z.; Ma, Y.; Liu, Y.; Xue, Y.; Li, Y.; Gao, X.; Wang, Y.; Chu, M. Advance in Identified Targets of Berberine. Frontiers in Pharmacology 2025, Volume 16-2025. https://doi.org/10.3389/fphar.2025.1500511
Karimi, S.; Heidari, F.; Amraei, S. Therapeutic Aspects of Berberine and Its Derivatives: Recent Advances and Challenges. Nano Micro Biosystems 2023, 2 (1), 42–47. https://doi.org/10.22034/nmbj.2023.390449.1018
Kong, Y.; Li, L.; Zhao, L.-G.; Yu, P.; Li, D.-D. A Patent Review of Berberine and Its Derivatives with Various Pharmacological Activities (2016–2020). Expert Opinion on Therapeutic Patents 2022, 32 (2), 211–223. https://doi.org/10.1080/13543776.2021.1974001 (to cover the patent literature).
Filli, M. S.; Ibrahim, A. A.; Kesse, S.; Aquib, M.; Boakye-Yiadom, K. O.; Farooq, M. A.; Raza, F.; Zhang, Y.; Wang, B. Synthetic Berberine Derivatives as Potential New Drugs. Brazilian Journal of Pharmaceutical Sciences 2022, 58. https://doi.org/10.1590/s2175-97902020000318835
Patel, P. A Bird’s Eye View on a Therapeutically ‘Wonder Molecule’: Berberine. Phytomedicine Plus 2021, 1 (3), 100070. https://doi.org/10.1016/j.phyplu.2021.100070 (this wide review considers also the antibacterial properties of berberine)
Wang, K.; Feng, X.; Chai, L.; Cao, S.; Qiu, F. The Metabolism of Berberine and Its Contribution to the Pharmacological Effects. Drug Metabolism Reviews 2017, 49 (2), 139–157. https://doi.org/10.1080/03602532.2017.1306544 (not only does this review include some papers dealing with the antimicrobial activities of berberine, but also discusses the main bioavailability issues that seriously hamper the use of this drug as a medicine).
Wang, J.; Tang, J.; Yu, L.-F.; Li, J.; Li, J.-Y. Berberine Analogues: Progress Towards Versatile Applications. HETEROCYCLES 2015, 91, 2233. https://doi.org/10.3987/REV-15-825.
We would like to thank the Reviewer for directing the enhancement of our Article. We have cited most of the proposed literature, excluding patent publications (we have excluded them in this Manuscript) and those to which we did not have access. Based on the others, we added the following section at the end of the Introduction:
„The reviews available in the literature focus primarily on the very important topic of overcoming BRB bioavailability. They extensively discuss its metabolic profile [41], physicochemical, pharmacological and toxicological properties [42]. They focus on the molecular action targets of this compound [43] and the impact of structural modifications to increase therapeutic effects, including improved bioavailability and absorption [44]. In addition, they discuss in detail the positive and negative side of the formulation of BRB and its derivatives [45]. However, they do not analyze in detail the antimicrobial activity of BRB and the attractive combinations that potentiate DDI (drug-drug interactions). After analyzing these reviews, there was a need to encapsulate the subject into a whole expanding the area of the effect of this isoquinoline alkaloid on the bacterial cell. Therefore, the main objective of this review is to provide a comprehensive overview of the antimicrobial mechanisms of action of this compound, which position it as a promising candidate for overcoming the growing challenge of antibiotic resistance.”
- The major ADMET issues in berberine as a medicine were not critically addressed.
We would like to thank you for this comment. Our Review does not directly address this topic, but we agree with the Reviewer that this problem should be highlighted. Accordingly, we have discreetly added single sentences indicating this problem.
„. At the same time, BRB, due to its weak absorption from the gastrointestinal tract and high toxic dose (41.6 g/kg), is a safe compound for clinical use [40] and particularly worthy of attention.”
” They are interesting from the point of view of antimicrobial activity and, in our opinion, worth special attention. This is because it seems that solving the problem of bioavailability and weak permeability of BRB using nanoparticles, phytosomes or BRB derivatives are the key to overcoming multidrug resistance.”
- The following paragraphs appear as written by AI:
It is difficult for us to respond to this comment, but we would like to point out that AI did not write a single part of this Manuscript. Yes, we occasionally asked AI's opinion, but this was to learn about its general approach. Moreover, this was for completely different parts than those indicated by the Reviewer. It is puzzling to us. Nevertheless, in order to avoid any misunderstanding and slander, we changed the parts indicated by the Reviewer.
Abstract: The increasing resistance to antibiotics poses a growing threat to public health 12 worldwide. The high prevalence of multidrug-resistant (MDR) strains, particularly 13 among Enterobacteriaceae and non-fermenting bacilli (Pseudomonas aeruginosa, Acinetobacter 14 baumannii) in intensive care units, severely limits treatment options. In response, 15 researchers are prioritizing the search for new generations of pharmaceuticals with 16 unique mechanisms of action
We changed the text to: ” In recent years, one of the most important issues in public health is a rapid growth of antibiotic resistance among pathogens. Multidrug-resistant (MDR) strains (mainly Enterobacteriaceae and non-fermenting bacilli) cause severe infections, against which commonly used pharmaceuticals are ineffective. Therefore there is an urgent need for new treatment options and drugs with innovative mechanisms of action.”
Research in this area is progressing rapidly. In recent years, promising strategies have emerged that specifically address this challenge. Notably, certain compounds have been developed that demonstrate the capacity to overcome traditional bacterial defenses. One such example is fabimycin—a rationally designed antibiotic that targets a key 87 enzyme involved in fatty acid biosynthesis in Gram-negative bacteria. This compound not 88 only exhibits enhanced permeability across the outer membrane but also circumvents 89 resistance mechanisms related to efflux pump activity [27,28].
We changed the text to: „Recent progress in research allowed the development of new drugs, invented in order to overcome bacterial mechanisms of antibiotic resistance. Fabimycin can serve as an example. It inhibits FabI, a rate-determining step catalyzing enzyme, which is involved in fatty acid synthesis. Its ability to accumulate inside targeted bacteria facilitates its effectiveness against resistant Gram-negative bacteria, which was proved in mouse infection model”.
In conclusion, BRB exhibits a range of antibacterial mechanisms distinct from 622 conventional agents, making it a strong candidate as a lead compound in combating 623 antimicrobial resistance. Further studies focused on its delivery methods and formulation 624 will be crucial in overcoming its potential toxicity to the human host. Such efforts may 625 offer a new therapeutic paradigm for managing infections caused by multidrug-resistant 626 pathogens, while also providing a robust platform for the design of next-generation 627 antimicrobials capable of circumventing existing resistance mechanisms.
We changed the text to: „In conclusion, BRB antimicrobial activity involves diverse mechanisms, including disruption of efflux pumps, suppression of QS and mitosis, inhibition of ?-lactamases. Those properties are stating its potential as a future treatment option in infections caused by antibiotic resistant bacteria. However, further research is needed to confirm efficiency and safety in humans, assess pharmacological properties and invent suitable forms for medical use. BRB can also serve as a starting point for development of even more useful derivatives or compounds with similar activity.”
They should be rewritten without the help of AI.
- Several grammar or syntactic errors, as well as typos, should be easily amended. An exemplary list is reported below.
Abstract, line 8: the hyphen after ‘alkaloid’ should be left out.
Introduction, page 2, line 13: ‘roduction of’ should be replaced with ‘acting as’.
We would like to thank you for bringing this to our attention. It has been changed.
Introduction, page 2, line 18: ‘interestingly’ is redundant and should be left out.
We would like to thank you for bringing this to our attention. It has been left out.
Introduction, page 2, lines 22-23: the end of the sentence (‘bacteria are ahead of us by several or even a dozen steps’) is not scientifically sound.
We would like to thank you for bringing this to our attention. It has been changed on: „the rate is still far too slow to overcome the development of bacterial resistance to antibiotics”.
Introduction, page 2, line 50: the indicated hydrogen should be typed in italics.
We would like to thank you for bringing this to our attention. It has been changed
Section 2, page 4, line 1: probably, ‘of’ should be placed between ‘study’ and ‘Li’.
We would like to thank you for bringing this to our attention. It has been added
Section 2, page 4, line 11: the article ‘The’ is missing before ‘Alkaloid’.
We would like to thank you for bringing this to our attention. It has been added
Section 2, page 4, line 13: a space should separate ‘24’ from ‘h’.
We would like to thank you for bringing this to our attention. It has been added
Section 2, page 4, line 18: the sign minus should be given as emdash (three times).
We would like to thank you for bringing this to our attention. It has been changed
Section 2, page 4, subsection 2.2, line 10: the sign minus should be given as emdash.
We would like to thank you for bringing this to our attention. It has been changed
Section 2, page 4, subsection 2.2, line 29: ‘g/mL’ or ‘microg/mL’?
We would like to thank you for pointing out this detail. However, the expression refers to the concentration of the compound, not the density of the test strain, so we have remained with “g/mL”
Section 2, page 6, subsection Enterobacteriaceae, line 3: I think that ‘the’ should be preferred to ‘an’.
We agree with the comment, ‘the’ has been added.
Section 3, subsection 3.1, page 7, line 4: why ‘e.g.’?
We would like to thank you for your attention. We have replaced with: ‘i.e.’
Section 3, subsection 3.1, page 7, line 18: the hyphen is not needed.
We would like to thank the reviewer for his insightful analysis. We wanted to stay with a uniform description. We used the notation ‘sub-MIC’ and, similarly, ‘sub-inhibitors’.
Section 3, subsection 3.3, page 8, line 15: the name ‘furanone’ should be followed by ‘C30’.
We would like to thank you for bringing this to our attention. It has been added
Section 3, subsection 3.4, page 8, line 1: ‘bacteria’ or ‘bacterium’?
We would like to thank you for bringing this to our attention. It has been changed.
Section 3, subsection 3.6, page 9, second paragraph, line 1: ‘contradictive’ or ‘contradictory’?
We would like to thank you for bringing this to our attention. It has been changed.
Section 3, subsection 3.6, page 9, second paragraph, line 1: ‘comes’ or ‘came’?
We would like to thank you for bringing this to our attention. It has been changed.
Section 3, subsection 3.6, page 9, second paragraph, line 15: ‘S. aureus’ should be typed in italics.
We would like to thank you for bringing this to our attention. It has been changed.
Section 3, page 10: the paragraph preceding Fig. 3 is redundant and may be left out.
Figure 3 is out of place and adds nothing: I suggest its removal.
We would like to thank you for your attention. However, in our opinion, this figure clearly illustrates the link between biofilm formation and the QS signaling we described. In our opinion, the reader has the transparency of the contribution of the compound we describe at different stages of this long-term process, which is important for clinical application. However, if the reviewer is not convinced then we will remove it in the next review step.
Section 4, subsection 4.1, page 10, line 5: ‘cirque’ or ‘cycle’?
We would like to thank you for bringing this to our attention. It has been changed
Section 4, subsection 4.3, page 11, line 2: ‘pure’ is redundant.
We would like to thank you for bringing this to our attention. It has been removed
Section 4, the second paragraph: The captions for the Figures and Tables make no sense there.
We would like to thank you for this comment. However, we must admit that we do not understand it. In Chapter 4, there is 1 Figure and its description is consistent with what it represents Could we ask you to concretize what exactly does not make sense? We will try to address this comment to the best of our ability.
We would like to thank the Reviewer very much for pointing out the weaknesses of our Review. We believe that after the corrections made, its value has increased and it is suitable for publication. We invite you to read the revised version of the Manuscript. We hope it will meet with your appreciation and you will be satisfied with the corrections we have made.
Best regards
Anna Duda-Madej

Reviewer 2 Report
Comments and Suggestions for Authors
The article "Exploring the Role of Berberine as a Molecular Disruptor in Antimicrobial Strategies" is well written and should be published in Pharmaceuticals. Some minor corrections are still needed, such as:
- the italicization of bacterial strains, such as S. Typhimurium (Table 1), should be checked throughout the article; row 331, etc.
- there are a lot of abbreviations and it might be easier for the reader if there was a list of abbreviations at the end of the article.
- it is good to give the exact name in the text in the case of some substances mentioned, such as,
-Br, ethyl bromide (row 218, 274).
-EDTA, Ethylenediaminetetraacetic acid (row 248).
-Col, Colistin (polymyxin E) (row 250...).
-COL-NS, colistin-nonsusceptible (row 249).
-3-oxo-C12-HSL, N-3-Oxo-Dodecanoyl-L-Homoserine Lactone (row 304).
-C4-HSL, N-Butanoyl-L-homoserine lactone (row 304)... etc
Author Response
Dear Reviewer,
Thank you very much for studying the topics undertaken in our Article and for your detailed review. In correcting our Review, we followed all your recommendations, for which we are very thankful. Our corrections are presented below:
- the italicization of bacterial strains, such as S. Typhimurium (Table 1), should be checked throughout the article; row 331, etc.
We would like to thank you very much for this comment. However, according to the latest guidelines, we write species names in lowercase and italics, while we write serotype names in uppercase and without italics. Below are articles on the nomenclature change of the genus Salmonella:
https://pmc.ncbi.nlm.nih.gov/articles/PMC86943/
https://pmc.ncbi.nlm.nih.gov/articles/PMC8021552/
- there are a lot of abbreviations and it might be easier for the reader if there was a list of abbreviations at the end of the article.
We would like to thank you for this good point. For clarity and easier reception of our Manuscript, we have included a list of abbreviations in alphabetical order at the end.
- it is good to give the exact name in the text in the case of some substances mentioned, such as,
-Br, ethyl bromide (row 218, 274).
-EDTA, Ethylenediaminetetraacetic acid (row 248).
-Col, Colistin (polymyxin E) (row 250...).
-COL-NS, colistin-nonsusceptible (row 249).
-3-oxo-C12-HSL, N-3-Oxo-Dodecanoyl-L-Homoserine Lactone (row 304)
-C4-HSL, N-Butanoyl-L-homoserine lactone (row 304)... etc
We would like to thank you for this good point. The abbreviation expansions of the compounds mentioned in our Article were added the first time they were used.
We would like to thank the Reviewer very much for pointing out the weaknesses of our Review. We believe that after the corrections made, its value has increased and it is suitable for publication.
Best regards
Anna Duda-Madej

Reviewer 3 Report
Comments and Suggestions for Authors
In Pharmaceuticals-3693032, Duda-Madej et al. discuss the role of berberine as a molecular disruptor in antimicrobial strategies. The topic of this review is interesting and aligns well with the scope of Pharmaceuticals. The reviewer believes the manuscript may be considered for acceptance after minor revisions.
(1) Currently, berberine is primarily administered orally and used for gastrointestinal conditions. It is well known for its poor membrane permeability and low oral bioavailability. If berberine is to be developed for systemic antibacterial applications, what would be the intended route of administration?
(2) Due to its poor systemic absorption, berberine is generally considered safe in clinical use. However, if administered systemically, are there any safety concerns that need to be addressed?
(3) Berberine is not particularly potent as an antimicrobial agent. Thus, higher doses may be required for systemic antibacterial efficacy, which may raise toxicity and tolerability issues. Could the authors comment on this?
(4) Could berberine potentially cause drug–drug interactions (DDIs) with other antibacterial agents when administered systemically?
Author Response
Dear Reviewer,
Thank you very much for studying the topics undertaken in our Article and for your detailed review. In correcting our Review, we followed all your recommendations, for which we are very thankful. Our corrections are presented below:
(1) Currently, berberine is primarily administered orally and used for gastrointestinal conditions. It is well known for its poor membrane permeability and low oral bioavailability. If berberine is to be developed for systemic antibacterial applications, what would be the intended route of administration?
Berberine has been studied intensively by scientists in recent years. It exhibits a wide range of activities. However, due to its low oral bioavailability and poor mucosal permeability, there is a need to focus future research on methods of its delivery. Promising forms are not only nanoparticles or phytosomes, but also berberine derivatives. Indeed, berberine hydrochloride has been shown to have a higher bioavailability than the pure compound, and is therefore more easily absorbed by the ogranism. We would like to thank the reviewer for this comment. We briefly addressed this topic at the end of the Introduction by adding the following sentences:
„They are interesting from the point of view of antimicrobial activity and, in our opinion, worth special attention. This is because it seems that solving the problem of bioavailability and weak permeability of BRB using nanoparticles, phytosomes or BRB derivatives are the key to overcoming multidrug resistance.”
(2) Due to its poor systemic absorption, berberine is generally considered safe in clinical use. However, if administered systemically, are there any safety concerns that need to be addressed?
We would like to thank you for pointing out this problem. We have marked it by adding one sentence to the Introduction. It has the following wording:
„At the same time, BRB, due to its weak absorption from the gastrointestinal tract and high toxic dose (41.6 g/kg), is a safe compound for clinical use [40] and particularly worthy of attention.”
(3) Berberine is not particularly potent as an antimicrobial agent. Thus, higher doses may be required for systemic antibacterial efficacy, which may raise toxicity and tolerability issues. Could the authors comment on this?
We understand the reviewer's concerns; however, we would like to inform you that our studies (as yet unpublished) show that the MIC and MBC for berberine against various Gram-positive and Gram-negative strains (including those exhibiting a variety of resistance mechanisms) is far lower than the toxic dose quoted by us for this compound (41.6 g/kg). If an effective method of getting this compound to its target site of action were developed, it would provide a valuable obstacle in the fight against multidrug resistance.
(4) Could berberine potentially cause drug–drug interactions (DDIs) with other antibacterial agents when administered systemically?
Previous studies have included interactions of berberine with other antimicrobial drugs. It has been shown that berberine can lower the MICs of individual antibiotics showing a synergistic effect. In this Review, we address this issue in several places. E.g., lines 176-178 address the reduction of MICs in the presence of berberine for amikacin, tobramycin and gentamicin by as much as 2-8 orders of magnitude for P. aeruginosa. In addition, synergistic effects were also seen in the combination of this compound with imipenem (lines 207-210) against P. aeruginosa and clarithromycin (lines 284 - 287) against M. avium.
We would like to thank the Reviewer very much for pointing out the weaknesses of our Review. We believe that after the corrections made, its value has increased and it is suitable for publication.
Best regards
Anna Duda-Madej

Round 2
Reviewer 1 Report
Comments and Suggestions for Authors
Dear Authors, I would like to congratulate you on your effort, which has surely significantly improved the quality of your paper. I think that it is suitable for publication. However, let me explain two indications of mine which, admittedly, were rather obscure in my first formulation.
Section 2, page 4, subsection 2.2, line 29: ‘g/mL’ or ‘microg/mL’?
I checked the referred paper, and it seemed to me that the concentration of the drug used in the paper was mg/mL, not g/mL. On the other hand, don't you think that g/mL would indicate an extremely low (and so useless) antibacterial activity?
Section 4, the second paragraph: The captions for the Figures and Tables make no sense there.
I intended to state that this part of the Section sounds just like a repetition of what the reader has found as captions under the corresponding figures and tables. I do not think that this information should be reported in Materials and Methods... maybe I am wrong.